# Age-related changes in tau and autophagy in human brain in the absence of neurodegeneration

Shreyasi Chatterjee[1,2☯], Megan Sealey[1☯], Eva Ruiz[1], Chrysia M. Pegasiou[1,3], Keeley Brookes[2], Sam Green[1], Anna Crisford[1], Michael Duque-Vasquez[1], Emma Luckett[1,4], Rebecca Robertson[1], Philippa Richardson[1], Girish Vajramani[5], Paul Grundy[5], Diederik Bulters[5,6], Christopher Proud[1,7], Mariana Vargas-Caballero[1]*, Amritpal Mudher[1]*

1 School of Biological Sciences, University of Southampton, Southampton, United Kingdom, 2 Department of Biochemistry, School of Science and Technology, Nottingham Trent University, Nottingham, United Kingdom, 3 School of Life Sciences, University of Sussex, Brighton, United Kingdom, 4 Department of Neuroscience, KU Leuven, Leuven, Belgium, 5 Department of Neurosurgery, Wessex Neurological Centre, University Hospital Southampton, Southampton, United Kingdom, 6 Faculty of Medicine, Clinical Neurosciences, Clinical and Experimental Sciences, University of Southampton, Southampton, United Kingdom, 7 Lifelong Health, South Australian Health and Medical Research Institute, SAHMRI, and School of Biological Sciences, University of Adelaide, Adelaide, Australia

☯ These authors contributed equally to this work.
* A.Mudher@soton.ac.uk (AM); M.Vargas-Caballero@soton.ac.uk (MV-C)

**Data Availability Statement:** All relevant data are within the manuscript and its Supporting Information files.

## Abstract

Tau becomes abnormally hyper-phosphorylated and aggregated in tauopathies like Alzheimers disease (AD). As age is the greatest risk factor for developing AD, it is important to understand how tau protein itself, and the pathways implicated in its turnover, change during aging. We investigated age-related changes in total and phosphorylated tau in brain samples from two cohorts of cognitively normal individuals spanning 19–74 years, without overt neurodegeneration. One cohort utilised resected tissue and the other used post-mortem tissue. Total soluble tau levels declined with age in both cohorts. Phosphorylated tau was undetectable in the post-mortem tissue but was clearly evident in the resected tissue and did not undergo significant age-related change. To ascertain if the decline in soluble tau was correlated with age-related changes in autophagy, three markers of autophagy were tested but only two appeared to increase with age and the third was unchanged. This implies that in individuals who do not develop neurodegeneration, there is an age-related reduction in soluble tau which could potentially be due to age-related changes in autophagy. Thus, to explore how an age-related increase in autophagy might influence tau-mediated dysfunctions in vivo, autophagy was enhanced in a Drosophila model and all age-related tau phenotypes were significantly ameliorated. These data shed light on age-related physiological changes in proteins implicated in AD and highlights the need to study pathways that may be responsible for these changes. It also demonstrates the therapeutic potential of interventions that upregulate turnover of aggregate-prone proteins during aging.

**Funding:** AM Gerald Kerkut Trust https://www.
kerkut-trust.org.uk/ Alzheimer's Research UK
https://www.alzheimersresearchuk.org/ No. The
funders had no role in study design, data collection
and analysis, decision to publish, or preparation of
the manuscript.

**Competing interests:** The authors have declared
that no competing interests exist.

## Introduction

Alzheimer's disease (AD) is the most common cause of dementia in the elderly, and with a growing population of older people, it is becoming increasingly more prevalent. Age is one of the greatest risk factors for developing AD, with incidence increasing from 1 in 14 above that age of 65 to 1 in 6 at age >80 (Alzheimer's Association report 2019). This clearly demonstrates that age is a significant risk factor for developing this condition.

Since many of the neuropathological hallmarks of AD and the clinical features that characterise it, such as declining working memory, short-term recall and processing speed, are also observed in normal aging [1–3], it has been proposed that these age-related changes are a continuum of AD [4]. However, the relationship between aging and AD is complex and interwoven, and though there are many areas of overlap [5], it is clear that AD is not an inevitable part of the physiological CNS aging process [6]. It therefore becomes imperative to understand what about the aging brain makes it vulnerable to AD compared to the younger brain; to understand the physiological changes that occur in aging brains and the possible mechanisms by which deviation from the normal aging process occurs and leads to pathology and neurodegenerative disease. This topic has been the subject of numerous reviews, each discussing the many theories that have been proposed to explain how aging confers risk of AD and other age-related proteinopathies [5,7]. One area of interest in this regard is the role of proteostatic pathways such as autophagy. Both macro-autophagy and chaperone-mediated autophagy are implicated in the turnover of tau, the microtubule associated protein that constitutes the neurofibrillary tangles, one of the two defining pathological hallmarks of AD [8–13]. One view is that there is a functional decline in autophagic clearance capabilities in the brains of individuals who are affected by neurodegenerative proteinopathies, and this manifests in the accumulation of misfolded protein aggregates [14,15]. Many mechanisms have been proposed to account for this functional decline. A large body of evidence suggests that autophagy is activated as a reactive response to misfolded protein formation at early stages of disease, but that the pathogenic proteins have a toxic effect on autophagy, compromising it such that it eventually becomes overwhelmed in proteinopathies [16]. Another view is that with increasing age, there is reduced autophagy and this is conducive to promoting aggregation and accumulation of misfolded proteins in the aged brain [17]. While many studies have demonstrated impaired autophagy in the brains of neurodegenerative disease patients, there are relatively few studies investigating normal physiological changes in basal autophagy with increasing age in the absence of neurodegeneration [17]. This is clearly an area with controversial findings and merits further investigation like the study we conducted.

In a similar vein, very few studies have shown how proteins like tau, which accumulate and misfold in Tauopathies and are believed to be substrates for autophagy, actually change physiologically in the human brain during healthy aging without the confounding influences of underlying neurodegeneration. The only study that assessed this in human brain was published over 20 years ago and reported a decrease in total soluble tau levels [18]. Though a mechanism was not demonstrated, that study nonetheless showed that the decrease in soluble tau was not due to a reciprocal age-related increase in insoluble tau.

The current study addresses these questions by investigating physiological age-related changes in both tau and autophagy in human brain samples taken from young and old subjects who did not have any neurodegenerative disease. In agreement with the study by Mukaetova-Ladinska et al [18], our findings also suggest that with increasing age, total soluble tau levels decline. Strengthening that previous finding, we demonstrate that this reduction in tau is seen in resected human brain tissue as well as post-mortem brain, confirming that it is not due to a post-mortem artefact. We also investigated whether hyperphosphorylated tau levels indicative

of AD pathology were altered in these non-AD cohorts and found no immunoreactivity in the post-mortem brain tissues. We then go on to investigate if this change in total soluble tau could be explained by an age-related change in autophagy. Though we do demonstrate an increase in expression of some markers of autophagy in older brains, others are unchanged. This data implies that an age-related increase in autophagy could influence total soluble tau levels and protect against tau-mediated neurodegeneration. To test this, a *Drosophila* model was employed in which co-expression of a pro-autophagic gene was found to significantly improve age-related tau phenotypes.

## Materials and methods

### Human tissue samples

Samples of frozen cortical sections from post-mortem human brains (six 20–33 year old brains and seven 70–74 year old brains) were obtained from Sudden Death Brain Bank (Edinburgh, United Kingdom, cortical area: superior temporal gyrus, case numbers included: SD023/08, SD041/05, SD027/06, SD034/08, SD030/09, SD001/11, SD008/12, SD020/06, SD053/14, SD025/17, SD024/17, SD012/17, SD008/17). No pathological diagnosis of Alzheimer's Disease or any other pathology was reported in these tissues.

Resected human Brain tissue was obtained during neurosurgical procedures at the Wessex Neurological Centre at University Hospital Southampton and the samples were processed according to the Human Tissue Act 2004 with approval from the Faculty of Medicine Ethics Committee and the Southampton Research Biorepository following written informed consent (Study reference number SRB002/14). The patients were being treated for medial temporal epilepsy, cavernous or arteriovenous malformations or glioma (Table 1). Any surplus tissue that was removed for access to these lesions that would otherwise be discarded was retained and subsequently processed as described by Pegasiou et al. [19]. Following resection cortical tissue was immediately put into artificial cerebrospinal fluid (ACSF) (110 mM choline chloride, 26 mM $NaHCO_3$, 10 mM D-glucose, 116 mM $C_6H_7NaO_6$, 7 mM $MgCl_2$, 3.1 mM $C_3H_3NaO_3$, 2.5 mM KCl, 1.25 mM $NaH_2PO_4$ and 0.5 mM $CaCl_2$.) The tissues were collected from patients in the age group 19 to 70 years and age-matched with the post-mortem brain samples. Thirteen samples were used for our experiments. These tissues were snap frozen on dry ice within 10 min of removal from the patient.

Both post-mortem and resected tissue samples were then lysed in homogenization buffer (10 mM Tris Base, 150 mM NaCl, 0.05% Tween-20, 1mM Na orthovanadate, 10 mM NaF, 10 μM Staurosporine, 1X Protease Inhibitor cocktail and 1 mM Tyrophostin) on ice. The samples were spun briefly at 8000 x $g$ for 5 min and the pellets were discarded. A protein assay was conducted by the Bradford method to normalize the protein concentration. The samples were then boiled for in 2x SDS sample buffer at 95˚C for 5 min for subsequent Western Blotting analysis.

### Electrophysiology

Following previously published methods [20], slices were transferred to a submerged-style recording chamber and superfused with recording ACSF at a rate of ∼1.5 mL/min. Whole-cell voltage clamp recordings were performed using glass pipettes (~5 MΩ) pulled from borosilicate glass, yielding a series resistance of ~15 MΩ. Recordings were made at room temperature (21-25˚C) using K-gluconate-based intracellular solution, containing (in mM): 120 Potassium gluconate; 10 KCl; 10 Hepes; 0.3 GTP; 4 Mg-ATP; and pH was titrated to 7.25 with KOH. The final osmolarity was 285 mOsmol$^{-1}$. Neuronal excitability was tested by delivering square pulses of current in increments of 50 pA.

**Table 1. Case study details of the human post-mortem brain and surgically resected tissues.** Post-mortem brain tissues were obtained from the Sudden Death Brain Bank and divided into young (20 to 33 years) and old (70 to 74 years) cohorts. There was no significant difference in the post-mortem intervals between the young and old cohorts (p = 0.48). Age-matched resected tissues (19 to 70 years) were collected from the Southampton General Hospital. Neuropathological abnormalities were found in cases 0011 and 0020 but no neurodegenerative changes were detected in any of the human tissues.

| Case Number | Sex | Age | Diagnosis | Other Medical History | Brain area |
|---|---|---|---|---|---|
| SD023/08 | F | 24 | Small vessel disease | Not known | Temporal lobe |
| SD041/05 | M | 24 | No significant abnormality | Not known | Temporal lobe |
| SD027/06 | M | 25 | Cerebral oedema. Recent hypoxia. History of butane gas inhalation. | Not known | Temporal lobe |
| SD034/08 | M | 70 | No significant abnormality | Not known | Temporal lobe |
| SD030/09 | F | 71 | No significant abnormality | Not known | Temporal lobe |
| SD001/11 | M | 74 | No significant abnormality | Not known | Temporal lobe |
| SD008/12 | M | 25 | No neuropathological findings | Not known | Temporal Superior |
| SD020/06 | F | 20 | No significant abnormality | Not known | Temporal Superior |
| SD053/14 | M | 33 | No significant abnormalities | Not known | Superior temporal gyrus BA41/42 |
| SD025/17 | M | 73 | No significant abnormalities | Not known | Superior temporal gyrus BA41/42 |
| SD024/17 | M | 72 | No significant abnormalities | Not known | Superior temporal gyrus BA41/42 |
| SD012/17 | F | 71 | No significant abnormalities | Not known | Superior temporal gyrus BA41/42 |
| SD008/17 | F | 71 | No significant abnormalities | Not known | Superior temporal gyrus BA41/42 |

**Resected Tissue Case Data**

| Case Number | Sex | Age | Diagnosis | Other Medical History | Brain area |
|---|---|---|---|---|---|
| 0019 | M | 19 | Arteriovenous malformation | Not known | Left frontal lobe |
| 0026 | M | 27 | Mesial temporal DNET with signal changes in the hippocampus | Not known | Right anterior temporal lobe |
| 0014 | M | 32 | Hippocampal Sclerosis | Not known | Left anterior temporal lobe |
| 0023 | M | 35 | Cavernoma | Not known | Left frontal lobe |
| 0016 | F | 36 | Hippocampal sclerosis | Not known | Right anterior temporal lobe |
| 0028 | F | 38 | Epilepsy | Asthma | Right anterior temporal lobe |
| 0011 | F | 42 | Glioma | Not known | Left anterior temporal lobe |
| 0024 | F | 50 | Cavernous malformation | Not known | Right lateral temporal lobe |
| 0012 | M | 55 | Glioblastoma | Not known | Left parietal lobe |
| 0017 | F | 62 | Hippocampal Sclerosis | Depression, hypertension, gastroreflux disease | Right anterior temporal lobe |
| 0025 | F | 62 | Metastasis—Adenocarcinoma of the lung | Not known | Right frontal lobe |
| 0013 | F | 69 | High Grade Glioma | Not known | Right frontal lobe |
| 0020 | F | 70 | Arteriovenous malformation | Hysterectomy, urostomy, acute myocardial infarction, hypertension, polymyalgia rheumatic, hypercholesteromia, recurrent urinary tract infections | Right lateral temporal lobe (superior temporal gyrus) |

Note: F, female; M, male; DNET, dysembryoplastic neuroepithelial tumor.

## Electron microscopy

Sections of freshly resected cortex were placed into a fixative solution consisting of 3% glutaraldehyde and 4% formaldehyde in 0.1 M PIPES and stored at 4°C for a minimum of 1 h. Brain

samples, approx. 1 mm by 1 mm in size, were washed with 0.1 M PIPES and then incubated in 1% osmium tetroxide for 1 h. After further washes in 0.1 M PIPES, samples were incubated in 2% uranyl acetate for 20 min. Dehydration followed using increasing concentrations of ethanol. After a final incubation in acetonitrile for 10 min, samples were incubated overnight in 50:50 acetonitrile:TAAB resin. Samples were then incubated for 6 h in TAAB resin and were then left to polymerise for 24 h at $60^{\circ}$C. 90–100 nm sections were produced from the samples using the Reichert OM-U3 ultra-microtome and placed onto copper palladium grids. Samples were then placed onto droplets of lead citrate for 3–5 min for further contrast before being imaged at x87,000 magnification using an FEI Tecnai T12 transmission electron microscope.

## RT PCR

RNA from the tissue samples was isolated using an in-house developed methodology [21]. Briefly, the Covaris Cryo Prep system was employed to crush the tissue to increase the surface area for allowing efficient cell lysis in 1 ml Trizol (Invitrogen). This was followed by RNA extraction with the RNAeasy Minikit from QIAGEN using standard protocols. The RNA concentration was measured in the nanodrop at an absorbance of 260 nm and the samples were processed for RT PCR. Briefly, 1 ug of RNA was treated with RNAse-free DNase I at $37^{\circ}$C for 30 mins to remove genomic DNA contamination. Following this treatment the DNAse was inactivated at $65^{\circ}$C and RT-PCR was performed by using the Superscript IV One-step RT-PCR kit (Invitrogen) with Platinum Taq Polymerase. Single strand cDNA synthesis was followed by PCR using 1 ug of RNA. For amplification of the 5' half of the tau mRNAs tau1 fwd primer (5'-ATGGCTGAGCCCCGCCAGGAG-3') and tau 4 reverse primer (5'-CCCAGCTCT GGTGAACCTCCA-3') were used [22]. These primers amplify tau sequences including exon 2 and 3, exon 2 alone or without these exons. The PCR was performed using the following conditions, pre-denaturation ($94^{\circ}$C for 2mins) followed by 40 cycles of amplification (denaturation, $94^{\circ}$C for 15s, annealing $60^{\circ}$C for 30s; extension 720C for 60s) and a final extension of 720C for 5 min. Another RT PCR was performed with human-β actin forward (5'-CCTCGCC TTTGCCGATCC-3') and reverse (5'GGATCTTCATGAGGTAGTGAGTC-3') primers for normalization. Products of RT-PCR were analyzed in 2% Agarose gels in TAE.

## Western blotting

*Human brain samples*: 20 to 30 μg of the homogenates were separated on 10% SDS-PAGE gels and transferred to nitrocellulose membranes/PVDF membranes. After blocking the membranes in 5% BSA in 1X TBS, these were incubated overnight at $4^{\circ}$C with the following antibodies. Anti-human tau (Dako, 1:10,000), the phosphorylation specific anti-tau antibody Ser396/Ser404 (PHF1, 1:500) (a gift from Dr. Peter Davies, USA), AT8 (Invitrogen, 1:1000), phospho-Ser-262 (Thermofisher, 1:1000) Acetylated α−tubulin (Sigma Aldrich, 1:1000), Tyrosinated α−tubulin (Sigma Aldrich, 1:2000), Total α-tubulin (Sigma Aldrich, 1:1000), p62 (Millipore, 1:1000), LC3 (Abcam, 1:500), GAPDH (Abcam, 1:2500), Beclin1 (Abcam, 1:500), Cathepsin D (Abcam, 1:1000), Rabbit β-actin (Abcam, 1:3000) and Mouse β-actin (Abcam, 1:1000) were used as loading controls. Membranes were then incubated with secondary antibodies (Alexa Fluor 680 goat anti-mouse or Alexa Fluor 800 goat anti-rabbit) at 1:20,000 for one hour at room temperature. Antigens were visualised using an Odyssey scanner (Li-Cor Biosciences).

*Fly brain samples*: Adult flies were snap frozen in liquid nitrogen and fly heads were dissected. Ten fly heads were then homogenized in homogenization buffer in a ratio of 1 head: 10 μl buffer (150 mM NaCl, 50 mM MES, 1% Triton-X, protease inhibitor cocktail, 30 mM NaF, 40 mM 2-glycerophosphate, 20 mM Na-pyrophosphate, 3.5 mM Na orthovanadate, 1%

SDS, 10 μM Staurosporine). Samples were centrifuged at 3000 x *g* and pellets were discarded. 2x SDS sample buffer was added to each tube and boiled at 95°C for 5 min. Western Blotting was then done as described above.

## Fly stocks

*Elav*-GeneSwitch, *UAS*-hTau$^{0N3R}$, *UAS*-Atg1 were obtained from Bloomington *Drosophila* Stock Center, Indiana, US. Bigenic flies (*UAS*-hTau$^{0N3R}$/*UAS*-Atg1) were created using standard genetic recombination methods. The flies were raised and maintained at 25°C on SYA diet using standard protocols.

## RU486 treatment

A 100 mM stock solution of RU486 was prepared in 100% ethanol. Post eclosion, 0–3 day-old flies were moved to either food containing 200 μM RU486 or to control diet for biochemistry and behavioural studies; this was to ensure gene expression was switched on only in adult flies. Only male flies were used for our studies to remove confounding impact of egg laying in the climbing assay that was done in these animals.

## Climbing assay

The climbing assay was performed according to standard procedures developed in our laboratory [23]. Briefly, five cohorts of 50 flies were transferred to 50 ml measuring cylinders without anasthetisation and the distance climbed was recorded at 10 seconds after tapping down. The assays was repeated three times with 1 min rest between each trial and the mean was calculated.

## Statistics

GraphPad Prism 5.0 software was used for statistical analysis. Two-tailed t tests were used to analyse the difference between two groups. For multiple comparisons one-way or two-way ANOVA with Bonferroni's post-hoc correction for pairwise comparisons was used. Error bars depict standard error of the mean (SEM) as indicated in the figure legend. Statistical significance is depicted by the following n.s = not significant ($p > 0.05$), *$p < 0.05$, **$p < 0.01$, ***$p < 0.001$, ****$p < 0.0001$.

# Results

## Resected human brain tissue is physiologically normal

To study impact of aging on tau in the absence of neurodegeneration in human brain, total and phosphorylated levels were assessed in non-pathological cortical human brain tissues taken from young and old individuals assessed by routine neuropathological analysis and by the lack of inflammatory markers [19]. Importantly, as previous studies [24–26], including ones from our own lab [27], have demonstrated that the post-mortem (PM) interval influences the phosphorylation state of tau; our initial studies were conducted both in resected human tissue following collection in the operating theatre and rapid processing (no PM interval) as well as in human tissue from national brain banks (standard PM interval). Resected tissue was collected at the neurosurgery unit of University Hospital Southampton from a spectrum of surgical cases ranging from 19 to 70 years of age. Tissue was either quickly frozen and kept at -80°C or processed for electrophysiology or electron microscopy (see methods). As described recently [19], the tissue utilised for these studies was physiologically and morphologically normal unless otherwise indicated. The rapid collection permits the preservation of function and

ultrastructure. We are confident that the resected tissue that we have utilised in this study represents normal physiological tissue without confounding disease. This is illustrated in S1 Fig where an electrophysiological recording taken from one such case (S1A Fig) showing robust action potential firing as a response to depolarising current injection, together with an ultrastructural image showing pre and postsynaptic structures (S1B Fig). Further electrophysiological recordings on these resected human tissue are shown in Table 1 [19]. Since resected tissue is a precious and scarce resource, only the analyses of total and phosphorylated tau, which we know from previous findings are sensitive to PM delay [28], were conducted using this tissue.

## Age dependent decrease in total tau in non AD brain

Post-mortem cortical human brain tissues from individuals without AD or other neurodegenerative disorders were obtained from the Sudden Death Brain Bank (Centre for Clinical Brain Sciences, Edinburgh, UK). These tissues were divided into young (aged 20 to 33 years) and elderly cohorts (aged 70 to 74 years) and age-related alterations in the levels of total and phospho-tau were assessed by western blotting. The post mortem intervals in the young and old cohorts were not significantly different (p = 0.48). Similarly, age-matched cortical resected tissues were freshly collected from individuals ranging in age from 19 to 70 years who were undergoing surgery (Wessex Neurological Centre, University Hospital Southampton) but did not have neurodegenerative disease. As with the post-mortem tissue, the resected tissues were divided into young (aged 19 to 42 years) and old cohorts (aged older brains 50–70 years). Due to limited availability, the resected human tissue could not be segregated into very young and very old, in contrast to the PM tissue, where samples of desired ages could be requested from brain banks. Nonetheless, the segregation of the resected tissue into young and old brain groups was performed based on previous findings that cognitive decline begins from 45 years onwards [29].

A significant decrease (p<0.0001) in the levels of total tau was observed in the post-mortem brain samples from the older cohorts compared to the younger cohorts (Fig 1A i and iii). This significant reduction in total soluble tau levels with increasing age was also observed in age-matched resected tissues (p<0.05) (Fig 1B i and ii). Interestingly, a robust phospho-tau signal (immunoreactive to the PHF1 antibody) was evident in resected tissues (Fig 1B i), but no such signal was detected in the post-mortem tissues (Fig 1A i) possibly due to the post-mortem delay. We did not observe a significant age-dependent increase of PHF-tau signal in the resected tissues (Fig 1B iii). The observation of robust PHF-1 positive tau in non-AD brain is unusual as tau is not believed to be significantly phosphorylated in normal adult brain (i.e., in the absence of neurodegeneration). This demonstrates the advantage of comparing signals in resected and PM tissues alongside each other to identify changes that may otherwise of be lost due to tissue processing and PM delay. We also tested two other AD associated phospho-epitopes AT8 and p-262 in the post-mortem tissues. While no signal was detected with the AT8 antibody, p-262 elicited weak antibody response in the samples (S2A and S2B Fig).

Next, we analyzed if this decrease of tau was due to transcriptional repression. We isolated total RNA from the post-mortem brain tissues and performed RT-PCR with tau-specific primers. We detected no significant difference in the 2N (567 bp), 1N (480 bp) or 0N (393 bp) tau transcripts between the youngers vs older cohorts (S3A and S3B Fig) suggesting that this variation of tau was at the translational level.

In order to confirm that the decrease in total tau levels in the older cohorts is not due to an age-related loss of neurons, the level of the neuronal marker NeuN was assessed by western blotting. No significant difference in NeuN levels was observed between the younger and older cohorts (Fig 1A ii and iii). Thus, we can conclude that the age-dependent decrease in the total soluble tau levels is not due to neuronal loss.

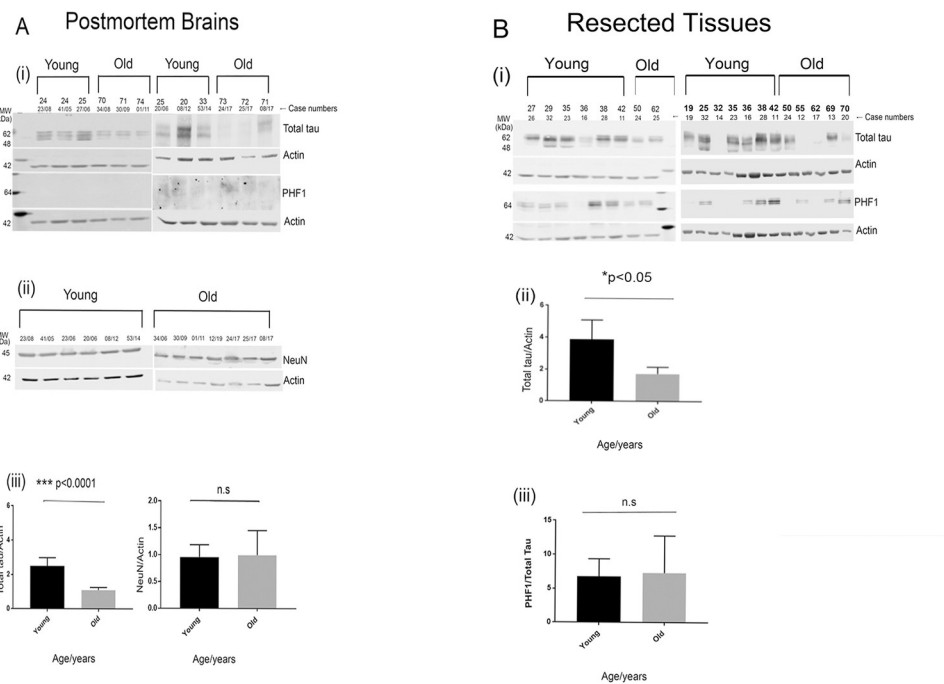

**Fig 1. Total tau levels are significantly decreased in the healthy older brains compared to the younger brains.**
Representative western blot images for total tau, PHF1 and Actin in post-mortem brains (A i) and quantification of total tau normalized to Actin (A iii). Representative western blot image for NeuN and Actin in post-mortem brains (A ii) and quantification of NeuN normalized to Actin (A iii). Representative western blot images for total tau, PHF1 and Actin in resected tissues (B i) and quantification of total tau normalized to Actin (B ii). Regression plot showing an increased trend of PHF1/Tau signal in the older cohorts compared to the younger group in resected tissue samples (B iii) In post-mortem samples n = 6 (older brains 70–74 years), n = 6 (younger brains 20–33 years), in resected tissue samples n = 13 (older brains 50–70 years and younger brains 19–42 years). ***p<0.0001, *p<0.05 by two-tailed unpaired t test. Data represent mean ± SEM (A iii and B ii). An increased level of PHF1 immunoreactivity was observed in older brains of resected tissues although it was not significant (B iii).

We next wanted to test whether the change in tau is correlated with an age-related alteration in tau-mediated function.

## Microtubule stability measures are similar in young and old brains

One of the primary functions of tau is the binding and polymerization of microtubules [30,31]. It has been shown that the hyperphosphorylated and PHF tau in AD brains can reduce the number and length of microtubules [32]. Further, it has also been shown that microtubule-stabilizing drugs are able to reverse axonal defects in animal models of tauopathy suggesting a direct connection between pathological tau and microtubule dysfunction [33]. In order to assess whether the loss of tau in the elderly cohorts compromises its function of polymerization and stabilization of neuronal microtubules, the levels of acetylated and tyrosinated alpha tubulin were assessed by western blotting. These post-translational modifications of tubulin are common indicators of microtubular integrity [34]. While acetylation is a marker for the stability of microtubules, tyrosination of tubulin has been shown to influence neuronal organization [34,35]. No difference in the levels of acetylated tubulin (Fig 2A i and ii) or tyrosinated tubulin (Fig 2B i and ii) was detected between the younger and elderly cohorts in the post-mortem brain samples. This implies that despite the decrease in cortical tau levels in healthy elderly cohorts, the stability of the microtubules in the same brain region is not affected.

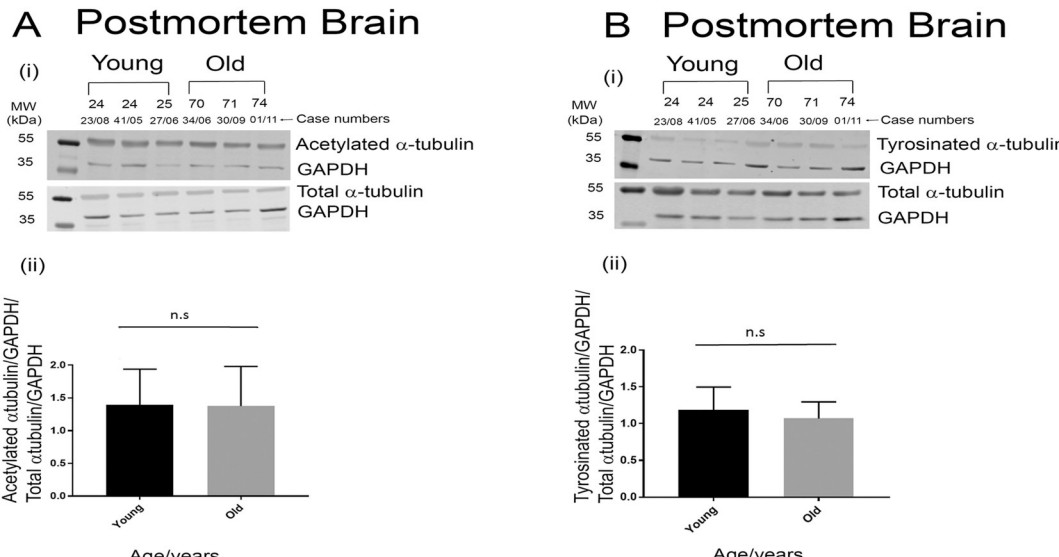

**Fig 2. The levels of total α-tubulin and its post-translational modifications are not altered with age in post-mortem brains.** Representative western blot images for acetylated α-tubulin, total α-tubulin and GAPDH in post-mortem brains (A i) and quantification of actetylated α-tubulin relative to total α-tubulin normalized to GAPDH (A ii). Representative western blot images for tyrosinated α-tubulin, total α-tubulin and GAPDH in post-mortem brains (B i) and quantification of tyrosinated α-tubulin relative to total α-Tubulin normalized to GAPDH (B ii). In post-mortem samples n = 6 (older brains) and n = 6 (younger brains). p values are not significant by two-tailed unpaired t test. Data represent mean ± SEM (A ii and B ii).

## Late autophagy markers are significantly increased in the post-mortem brains of the elderly cohorts compared to the younger cohorts

To investigate the potential cause for the age-related decline in total tau levels, markers of autophagy were studied in the young and old post-mortem brain samples. p62 is a classical autophagy marker, its function being the ubiquitination of a host of cellular proteins that are then targeted for degradation [36]. As it is destroyed during autophagy, accumulation of p62 implies an impairment of autophagy while a reduction of p62 is indicative of activated autophagic clearance. A significant decrease of p62 ($p < 0.05$) was evident in the post-mortem brains of the older cohorts compared to the younger cohorts (Fig 3A i and ii). As there is no significant difference in the PM delay in these samples which may be influence this readout, these results potentially point to increased markers of autophagy in the brains of the elderly cohorts.

To investigate this further, additional markers of autophagy were investigated. In mammalian systems, another classical marker for late-stage autophagosome formation is the lysosomal marker LC3-II, and an increase in this signifies an enhancement of autophagy [37]. In addition to decreased p62, a significant increase in the LC3-II/LC3-I ratio ($p < 0.05$) as well as an increase in total LC3-II levels was evident in the older post-mortem brains compared to the younger brain tissues (Fig 3B i-iii).

Since both p62 and LC3II/I ratios are markers of late-stage autophagy, we next investigated whether early stage markers were also altered with age. For this we assessed levels of Beclin 1, a key autophagy protein known to be decreased in AD brains [38]. Interestingly, we did not see a significant difference in the levels of this early autophagy marker–neither did this marker go up or down with age in our young and old non-AD cohorts (Fig 4 i and ii). Another important pathway for tau degradation; particularly in the AD brains is the lysosomal clearance pathway. Therefore, we assessed the levels of the lysosomal protease Cathepsin D in the younger and older non-AD cohorts. However, no significant difference was observed (S4 Fig).

## A Postmortem Brain

(i)
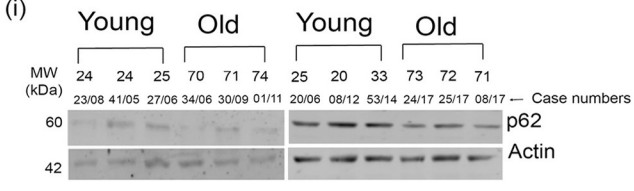

(ii)
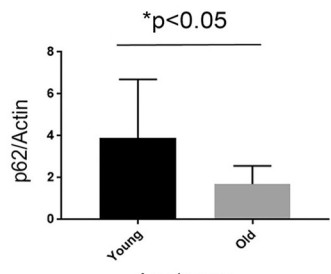

## B Postmortem Brain

(i)
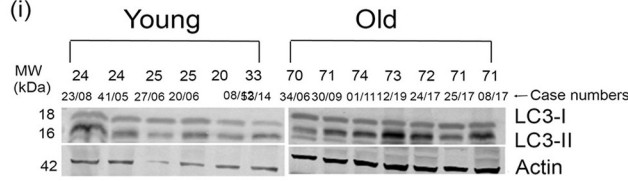

(ii)
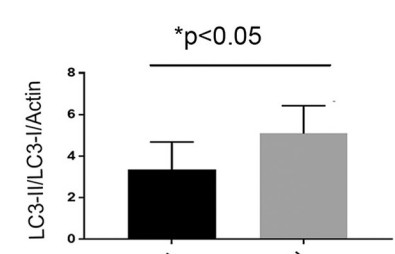

(iii)
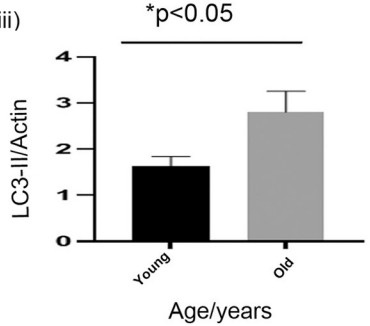

**Fig 3. Autophagy is significantly increased in the healthy older brains compared to the younger brains in post-mortem tissues.** Representative western blot images for p62 and Actin in post-mortem brains (A i) and quantification of p62 normalized to Actin (A ii). Representative western blot images for LC3 and Actin in post-mortem brains (B i) and quantification of LC3-II/LC3-I normalized to Actin (B ii) and LC3-II normalised to Actin (B iii). In post-mortem samples n = 7 (older brains), n = 6 (younger brains). *p<0.05, by two-tailed unpaired t test. Data represent mean ± SEM (A ii and B ii).

Taken together our results indicate that expression of some late-stage autophagy markers is increased in the older compared to the younger cohorts. However as the one early stage marker we examined appeared unaltered, it is not clear how overall autophagy changes with age, in the absence of neurodegeneration. Nonetheless it is clear that there is no age-related decline in autophagy as has been the consensus view.

### Age-related tau-induced behavioural changes are suppressed by activation of autophagy in a *Drosophila* model of Tauopathy

The data presented thus far show that in non-AD brains, soluble tau levels are decreased with age, whilst expression of some specific markers of autophagy is increased. One interpretation of this is that an age-related increase in autophagy is a normal physiological response to aging which likely prevents the accumulation of total tau and protects against formation of tau aggregates in non-demented elderly subjects. To test this hypothesis, we assessed the impact of inducing autophagy on some age-related tau phenotypes in a *Drosophila* model of Tauopathy

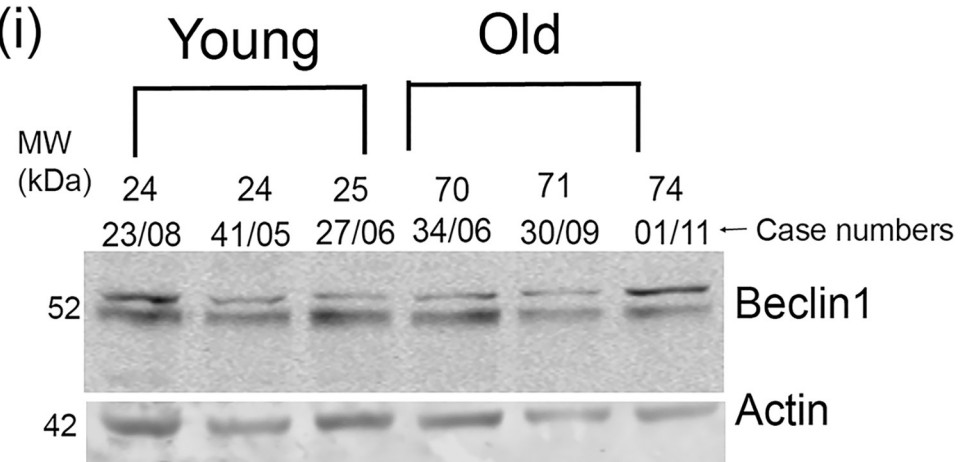

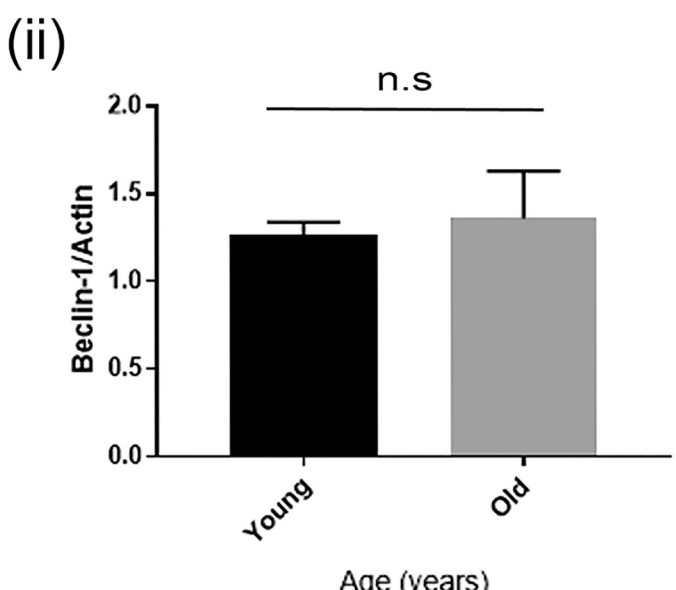

**Fig 4. Early autophagy gene Beclin 1 is not significantly increased in the older brains compared to the younger brains in post-mortem tissues.** Representative western blot images for Beclin 1 and Actin in post-mortem brains (i) and quantification of Beclin 1 normalized to Actin (ii). In post-mortem samples n = 7 (older brains), n = 6 (younger brains). p values are not significant by two-tailed unpaired t test. Data represent mean ± SEM.

in which hTau$^{0N3R}$ is expressed. 0N3R is one of the human Tau isoforms that has three micro-tubule binding repeats in the C-terminal region and no N-terminal repeats. In human brains the accumulation of the 0N3R isoform of tau is mainly causative of Pick's Disease but 3R tau is also important for AD pathology [39]. In this model, autophagy was induced by upregulation of the autophagy-specific kinase gene Atg1 [40]. To avoid developmental impact of inducing autophagy, pan-neuronal expression of both hTau$^{0N3R}$ and the autophagy gene manipulation was initiated in adults using the inducible GeneSwitch system in which the drug RU486 was fed to newly eclosed (emerged) adults to induce expression in adult flies which were then aged for a 4 week period. In parallel, the tau/Atg1 bigenics were similarly fed RU486 upon eclosion

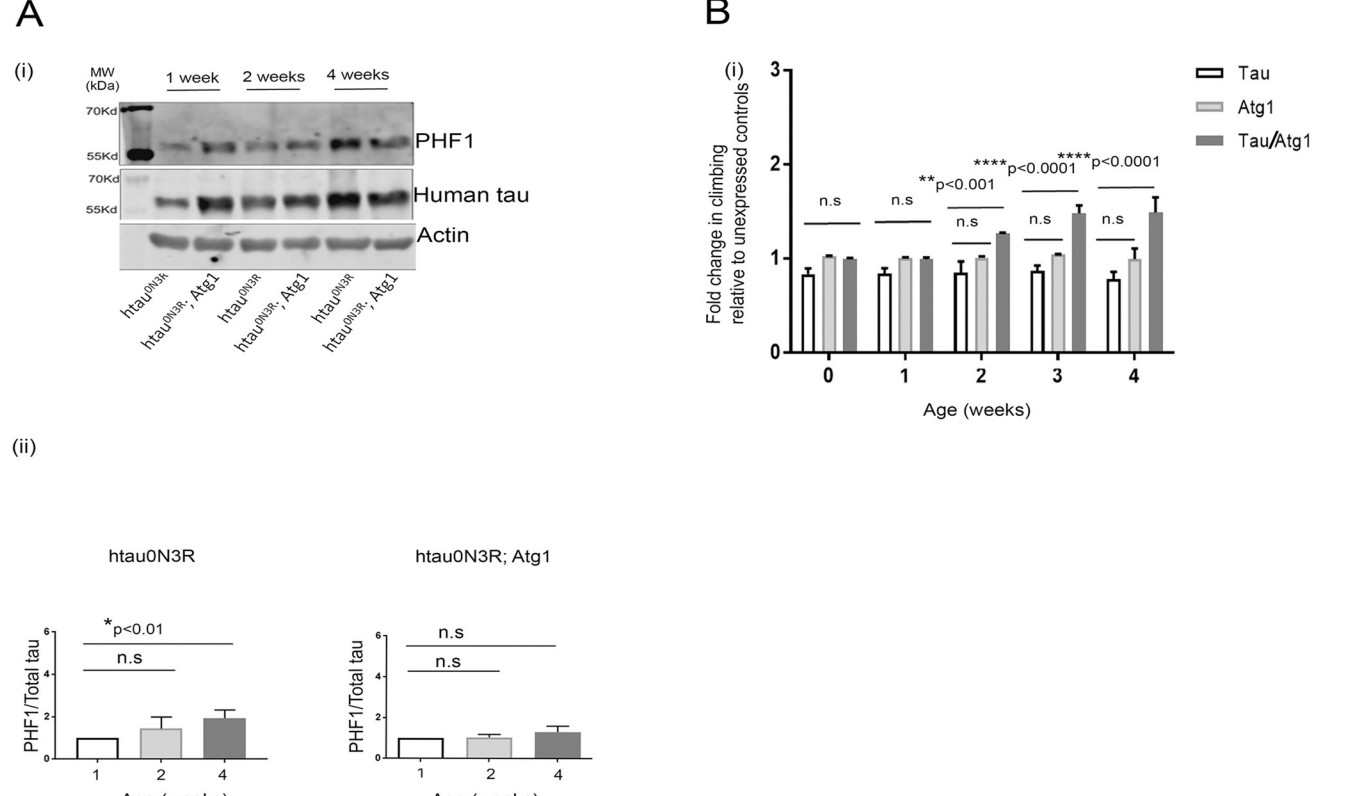

**Fig 5. Upregulation of autophagy decreases the age-related accumulation of hyperphosphorylated tau and ameliorates tau-induced locomotor defects in human Tau (0N3R) expressing transgenic flies.** Pan-neuronal expression of human 0N3R tau significantly increases the age-dependent accumulation of hyperphosphorylated tau that is restricted by coexpression of autophagy marker Atg1. Representative western blot images for PHF1, total tau and Actin (A i) and quantification of PHF1 relative to total tau normalized to Actin (A ii) in htau$^{0N3R}$ and htau$^{0N3R}$/Atg1 overexpressing *Drosophila* models respectively. *p<0.01 by one-way ANOVA with Bonferroni's correction. Data represent mean ± S.E.M. Expression of 03NR human tau pan-neuronally in adult flies induces locomotor deficits that are rescued by coexpression of Atg1 (B) Comparison of the climbing ability with age over a period of 4 weeks for hTau$^{0N3R}$, Atg1 and hTau$^{0N3R}$/Atg1 transgenics (n = 50). (2 way ANOVA; *p<0.05, **p<0.001, ****p<0.0001). Error bars are plotted as ± S.E.M. Genotypes: hTau$^{0N3R}$ = {w; *Elav*-Geneswitch/*UAS*-htau$^{0N3R}$}, Atg1 = {w; *Elav*-Geneswitch/+; *UAS*-Atg1/+}, hTau$^{0N3R}$/Atg1 = {w; *Elav*-Geneswitch /*UAS*-htau$^{0N3R}$; *UAS*-Atg1/+}, on an OreR background.

and aged for the same period of time. This treatment led to induction of expression of both hTau$^{0N3R}$ and autophagy (S5 Fig). With this mode of adult-onset hTau$^{0N3R}$ expression, a significant age-related increase (p<0.01) in phosphorylated tau was evident which was abrogated by co-expression of Atg1 (Fig 5A i/ii).

We have previously demonstrated that overexpression of hTau$^{0N3R}$ using the non-temporally sensitive *Elav*-GAL4 pan-neural driver causes an age-dependent defect in the climbing ability of the flies [13]. Such a reduction in climbing ability was beginning to emerge in the flies expressing hTau$^{0N3R}$ following RU486 consumption, though it was less pronounced over the 4-week period tested, possibly because, with the GeneSwitch method employed here, tau expression only begins upon eclosion, but is likely to begin several days before eclosion with the *Elav*-Gal4 expression system previously used (Fig 5B). Induction of autophagy by co-expression of Atg1 significantly (p<0.0001) improved the climbing behaviour of the hTau$^{0N3R}$ flies. In young flies (up to 1wk old) there was no difference in the climbing ability of flies expressing hTau$^{0N3R}$ alone when compared to hTau$^{0N3R}$ flies that co-expressed Atg1. However, as the hTau$^{0N3R}$/Atg1 bigenics started aging, their climbing ability was significantly rescued by coexpression of Atg1 compared to the age-matched hTau$^{0N3R}$ flies alone. After 2 weeks, the

climbing ability of the hTau$^{0N3R}$/Atg1 bigenic animals became significantly better (p<0.001) than that of hTau$^{0N3R}$ flies so that by 4 weeks their climbing was nearly 100% better (p<0.0001). This effect was tau-specific and not a non-specific effect of over-expressing Atg1 since there was no similar age-dependent improvement in the climbing ability of flies expressing Atg1 alone. These results show that induction of autophagy prevents the age-related accumulation of hyperphosphorylated tau and improves the climbing behaviour of tau-expressing flies in an age-related manner.

Curiously, the climbing behaviour of the hTau$^{0N3R}$;Atg1 bigenics was not just improved compared to those of either hTau$^{0N3R}$ or Atg1 alone flies, it was also better than the unexpressed bigenic controls. One potential explanation for this is that, as well as mediating a toxic effect, expression of hTau may have some other beneficial effect not evident in the unexpressed controls. One such effect may relate to endogenous *Drosophila* (dTau) as we have previously demonstrated that the two interact [41]. To explore this, the impact of hTau$^{0N3R}$ expression on dTau was examined. dTau levels were found to accumulate with age in wild-type flies but not in flies expressing hTau$^{0N3R}$ (S6 Fig). It is likely that, in the presence of hTau, endogenous dTau expression is reduced as a compensatory measure. Consequently any detrimental effects caused by dTau accumulation would not be evident in hTau expressing flies but would still be present in unexpressed controls [42,43].

## Discussion

Age is the biggest risk factor for Alzheimer's disease [44,45] and yet the mechanism(s) by which increasing age confers risk of disease are not well understood. In this study the impact of age on total and phosphorylated tau levels was studied in human brain tissue taken from young and old subjects, together with an analysis of age-related changes in autophagy, a pathway implicated in tau clearance [46]. With increasing age, a decrease in total tau levels and an alteration in two markers indicative of increased autophagy were evident in the absence of overt neurodegeneration. In order to test the impact of increased autophagy on human tau during aging, autophagy was genetically upregulated and tau-related behavioral changes were studied in young and old transgenic *Drosophila*. Co-expression of an autophagy gene, Atg1, with human tau ameliorated the accumulation of phosphorylated tau and significantly improved their climbing behaviour in an age-related manner in this model. Collectively these data suggest that age-related increases in autophagy may prevent the accumulation of tau and thus protect against tau-mediated pathogenic changes *in vivo*. It remains to be seen whether this is the mechanism by which elderly individuals who do not develop AD are protected and is worthy of further investigation. Similarly, whether it is a *deviation from this trait* that leads to compromised autophagy in some individuals as they age which then causes tau accumulation and downstream degeneration also needs further exploration (S7 Fig).

### Age-related changes in tau in non-AD brain

Aging is defined as a time-dependent deterioration of the physiological functions of our body. Whilst the biological process of aging is complex and not well defined, it has been established that age is the primary risk factor for Alzheimer's Disease (AD) with 90% of the AD cases occurring when the patients are 65 years or older (Alzheimer's Association 2019). There are many theories to explain how aging confers risk of developing Alzheimer's disease [5,6]. One view is that the formation of neuritic plaques and neurofibrillary tangles with increasing age [47] is a continuum of AD and that individuals with intermediate levels of these pathologies display mild cognitive symptoms whilst those with higher levels of pathology display full blown dementia [4]. Though attractive, this view is challenged by the finding that a significant number of individuals with neuritic

plaque and tangle pathology are not cognitively impaired [48,49]. This indicates that increased formation of such pathologies with increasing age cannot entirely explain how aging increases risk of developing dementia, and that other factors may also play a role [50].

Nonetheless, since there is a tight correlation between formation of neurofibrillary pathology and cognitive decline, it is worth investigating how tau proteins change in normal, non-demented individuals with increasing age. Whilst many studies have studied tau protein in brain and CSF from Alzheimer patients, and by default in most of these studies an analysis of age-matched controls is included, relatively few studies have assessed how tau protein itself changes with aging by directly comparing it in CNS tissues from young and old non-demented individuals. The data presented here demonstrates that total tau levels in cortical samples from older individuals are significantly reduced when compared to those found in younger individuals. This is likely to reflect an age-related decline in total soluble tau rather than a non-specific consequence of neuronal death during aging since the neuronal marker NeuN [51] does not change, indicating that there is no significant age-related neuronal loss in these cohorts, a phenomenon reported by others as well [5].

Our data cannot be attributed to brain-bank specific peculiarities in processing of tissues, as our samples were obtained from two distinct brain banks. Nor can it be attributed to age related changes in the tau mRNA levels. Additionally our data is in agreement with one other study that has similarly compared tau levels in young and old non-cognitively impaired human brain samples. They also showed that soluble total tau levels decreased with age in individuals ranging from 19–88 years, but were not able to identify a potential mechanism. Nonetheless they did show the reduction in soluble tau could not be explained by a correlative increase in insoluble tau suggesting that there must be another explanation for the age-related reduction of soluble tau that both they and we observe [18]. As well as this, one more recent observation that also supports this result showed immunohistochemical decreases in tau in retinal biopsies taken from older subjects compared to those from younger subjects [52]. Building on these findings, we show here that in the brain samples where soluble tau is decreased, there is also an age-related increase in expression of late-stage markers of autophagy, but it is unclear whether this contributes to the age-related reduction in soluble tau. To address this issue we investigated whether that the reduction of soluble tau levels in the older subjects could also be due to transcriptional repression of the tau-gene. Interestingly, we observed equal amounts of tau isoforms indicating that the decrease of total tau that we observe in older cohorts is not due to decreased transcription in these samples.

Even fewer studies have reported age-related changes in tau phosphorylation in non-demented individuals, let alone assess how these change with age. This is because the consensus view is that tau is not extensively phosphorylated in cortical regions of non-demented individuals who do not have overt pathology [53] with only a limited number of studies reporting some physiologically phosphorylated epitopes in non-AD brains [54,55]. In agreement with this, very little phosphorylation of tau was evident, at the disease epitopes PHF-1, AT8 and phospho-serine 262 epitopes that were examined in this study, in any of the cortical brain tissues taken from non-demented individuals where there was a post-mortem delay. However, phosphorylation at the PHF1 site was clearly evident in resected cortical tissues taken from both young and old non-demented individuals, and there was a trend for this to increase with age, though this was not significant.

One study published over 20 years ago made similar observations, showing that tau derived from biopsy non-AD brain tissue was more highly phosphorylated at several sites, thought to be primarily AD-specific, including PHF-1, AT8 and 12E8 and that phosphorylation at most of these sites was dramatically reduced if a post-mortem delay was artificially simulated, even if for 5–10 mins [24]. We have previously shown that this is also true for endogenously

phosphorylated rodent tau, where strong AT8 immunoreactivity is evident in hippocampal neurones (prominently localised to the somatodendritic compartment), which is reduced by nearly 75% when a post-mortem interval is simulated [27]. As well as a lack of PM delay, one cannot exclude the impact of one other difference between the resected tissue and the PM tissue which is that in the former the patients underwent anaesthesia, which has been shown to increase tau phosphorylation [56,57]. However, we have previously demonstrated that in rodent brain, even in the absence of anaesthesia, tau is significantly more phosphorylated when tissue is processed quickly without PM delay compared to when it is processed with a PM delay [27]. A post-mortem delay of several minutes is also created by default during trans-cardial perfusion fixation in rodent models of tauopathy, and this may also influence the phosphorylation status of any soluble tau species in those models. Collectively, these findings and our own strongly suggest that soluble tau is more phosphorylated in cortical tissues from non-demented individuals than is generally reported, but that the phosphorylation status is significantly reduced by the post-mortem delay. Phosphorylation status is adequately preserved in resected tissue, that though taken from individuals undergoing neurosurgery, is taken from brain regions that are physiologically and ultra-structurally normal as shown by us (in S1 Fig) and previously by others [12,15]. Studies in such tissues may be useful in future to assess if physiological phosphorylation of soluble tau increases with age, enhancing age-related risk of aggregation.

Tau aggregates composed of soluble and insoluble oligomers are one of the crucial factors driving the spread of tau pathology in the AD brain [58]. However, previous studies from Ladinska et al. has shown that the decrease in soluble tau observed in non-demented elderly cohorts does not correlate with the increase of insoluble tau in these brain samples [18]. Therefore, we did not investigate the levels of tau aggregation in these samples.

## Age-related changes in cytoskeletal integrity

Since microtubule stabilisation is a key physiological function of tau, one may hypothesize that the age-related reduction in total tau may manifest in reduced cytoskeletal integrity. To assess the impact of age-related reduction of total tau, on the neuronal cytoskeleton, acetylation and tyrosination of alpha-tubulin, two post-translational modifications that may indicate its stability, were probed [59–61]. Despite the significant reduction in total tau levels in the elderly cohorts compared to the younger cohorts, no change was observed in the acetylation or tyrosination levels of alpha-tubulin between the two groups. This implies that the loss of tau in the healthy control brains does not affect altered stability of microtubules with increasing age. It is possible that microtubule stability is only disrupted in the situation of tau hyperphosphorylation and profound neuronal loss that is observed in AD but not in aging control brains, or that age-related loss of tau is compensated for by an upregulation of other microtubule-associated proteins thereby preserving the neuronal cytoskeleton. This result is in disagreement with that of Cash et al. [31], who reported an age-related reduction in cytoskeletal integrity in cortical biopsy samples from non-AD patients. However, their cortical samples were taken from more elderly individuals than ours, and their age range spanned 2 decades, ranging from 62–80 years, whereas ours spanned 5 decades ranging from 24–74 years.

## Age-related increase in markers of autophagy and age-related decline in total tau

In order to examine the underlying biochemical pathways possibly responsible for the age-related reduction of total tau, we explored whether there were age-related changes in macroautophagy, a pathway implicated in tau turnover [62,63]. There is a lot of interest in protoeostatic pathways such as autophagy and the ubiquitin-proteasome system, as they have been shown to regulate

turnover of tau, and its accumulation in Tauopathies is postulated to arise due to an age-related functional decline in the efficiency of these pathways [64–66]. However, although there are numerous studies reporting autophagic induction in AD brains and comparing that to age-matched control brain [67], there are only limited reports comparing markers of autophagy in brains of young and old non-AD subjects. Two studies that have examined this suggest that macroautophagy decreases with increasing age in normal brain, as shown by an age-related down-regulation of several autophagy related genes [68] and Beclin 1, a protein essential for autophagy [69]. The data presented here suggests that the situation is more complicated and one cannot necessarily assume that there is an age-related reduction in autophagy as the consensus view suggests.

We show that two markers of late-stage autophagy are increased with age, the ratio of LC3 II/I, a sensitive marker of macroautophagy [70], with a corresponding reduction in p62 (which is destroyed during autophagy), thus implying greater autophagy [71], whilst one marker of early stage autophagy, Beclin 1 is unchanged. Indeed increased autophagy with advancing age has been reported in other tissues [72,73] and in choroid epithelial cells in brains of non-AD subjects [74]. This is further supported by some studies of aging in experimental models, where markers of autophagy including LC3 II/I ratios and pro-autophagic regulators like BAG-3 are higher in brains of old animals when compared to young animals [75]. However, there are conflicting findings suggesting age-related decline in autophagy in other experimental models [76]. Confusing the issue further, autophagy is believed to be upregulated with increasing age in long-lived animals and in healthy long-lived humans [77,78].

Next we analyzed Cathepsin D which is a lysosomal protease that has been found to be upregulated in AD [79]. In our study the levels of Cathepsin D were not significantly altered between the younger and the older cohorts in the post-mortem brain despite seeing a significant increase in some autophagy markers in the older cohorts. Although there is an intimate connection between Cathepsin D and autophagy, this has been mostly reported in the neurological disease conditions [80]. Given that these samples were collected from non-AD subjects there may not be detectable alterations in the levels of Cathepsin D.

The findings presented here and the conflicting reports from other studies on physiological age-related changes in autophagic capacity, very few of which were conducted using material from human brain, collectively imply that the situation is complex and requires further investigation [81]. Such investigations should also assess the cellular localisation of markers of autophagy, since some markers, like LC3-II, may appear to increase in conditions when autophagic flux is blocked, complicating interpretation of biochemical readouts [71]. Only then could a consensus be reached as to whether autophagic capacity changes with age, especially in human brain, and whether this *per se* confers higher risk of neurodegenerative proteinopathies in the elderly.

Whilst the evidence for changes in autophagy with age is confusing, it has consistently been shown by many that autophagic clearance can regulate tau turnover [20,81]. The age-related decline in soluble tau that we report may therefore occur at least in part due to the age-related increase in autophagy suggested by the increased markers of autophagy that is also evident in these brains. However further studies will be needed to confirm this and the role of lysosomal proteases in degrading tau in post-mortem non-demented AD brains. Furthermore, since autophagy deficiency leads to age-dependent neurodegeneration [62], it is possible that the lack of neurodegeneration in the brains of the older subjects in this study is a consequence of their healthy autophagic capacity. One may speculate that an age-related increase in effective autophagic capacity is a central component of healthy aging and protects healthy individuals from developing AD as they age (S7A Fig blue line). However, deviation from this, for example if the soluble tau (or amyloid beta) load overwhelms the autophagic system, could trigger the onset of AD in some individuals (S7A Fig, red line). Indeed, many studies have shown that

macroautophagy is impaired in AD brains [67] as well as in animal models of tauopathy [82]. An alternative hypothesis, based on the consensus view discussed earlier, is that a reduction in autophagic capacity is evident in all individuals with increasing age, and this is the reason why some individuals might develop AD (S7B Fig). Though the latter does not explain why only some aged indiviudals go on to develop proteineopathy. Another interesting study conducted by Kang et al, demonstrated that both normal and pathological tau is spread in the neurons by autophagy inducers [83]. Hence, further studies, using additional markers of autophagy in a larger cohort are needed to distinguish between these two scenarios.

## Upregulation of autophagy rescues age-related tau phenotypes

Our observation of increased markers of autophagy in aging human brains, evident in the face of decreased total tau levels and the absence of tau pathology, led us to hypothesize that induction of autophagy could ameliorate tau abnormalities in an age-dependent manner in a *Drosophila* model. To test this hypothesis, we assessed the impact of upregulating autophagy in a *Drosophila* model of Tauopathy in which we have previously described age-related tau phenotypes [22]. *Drosophila* is an ideal organism in which to test such hypotheses because it has been used for decades for genetically dissecting the pathways that underpin aging, autophagy and tau-mediated neurodegenerative diseases [84–86]. Indeed others have previously also shown that induction of autophagy in both rodent and *Drosophila* models of tauopathy reduces tau pathology and phenotypes [87–92] but the impact of aging has not been investigated.

In agreement with these studies, our data show that co-expression of Atg1, a key autophagy protein ameliorates the age-dependent increase in tau phosphorylation and significantly improves the climbing ability of hTau$^{0N3R}$ expressing flies, a behaviour we have previously shown is associated with tau hyperphosphorylation [23]. Interestingly, the beneficial effect of inducing autophagy was age-dependent, being more evident in older flies compared to younger flies. That increased autophagy is benefical for aging flies, as our data indicates, is supported by several other reports which demonstrate that both pharmacological and genetic upregulation of autophagy extends lifespan in invertebrate and vertebrate models of aging [81,90], which in some cases is accompanied by suppression of brain aging and a reduction in total tau [91,92].

The consistent conclusion that can be drawn from all such studies is that robust autophagic capacity may be key for preventing the age-related accumulation and thus toxicity of aggregate-prone proteins such as tau, thus enabling healthy brain aging and protecting against neurodegeneration. Our studies also add to the growing weight of evidence suggesting that the activation of autophagy could be an important therapeutic target for the treatment of tauopathies [93–97].

## Conclusions

Our data highlight the importance of studying the effect of aging on the normal physiological turnover of aggregate-prone proteins like tau as well as the mechanistic pathways that might regulate this in the absence of disease. It is especially important to undertake such studies using normal human brain tissue to identify the impact of age-related changes without the confounding influence of underlying neurodegenerative disease. This is particularly relevant in understanding how aging confers risk of proteinopathies such as Alzheimer's disease.

## Supporting information

**S1 Fig. Electrophysiological recordings and ultrastructure of cortical resected tissue.** Whole-cell patch clamp recording from a layer II-III human cortical pyramidal neuron

showing active voltage responses to current injection (A). Electron micrograph showing a cortical synapse. Scale Bar 200 nm (B).
(TIF)

**S2 Fig. Phospho-tau immunoreactivity (AT8 and phospho-262) in the post-mortem brain samples was low.** Representative Western Blot images for AT8, ps-262 and β-actin in post-mortem brains (A and B). There was no significant difference in the AT8 and ps-262 immunoreactivities between the younger vs older cohorts.
(TIF)

**S3 Fig. Electrophoresis of RT-PCR amplification products of tau mRNA show no significant difference between the younger and the older cohorts of the post-mortem brain samples.** The three PCR products correspond to the 2N, 1N and 0N forms of tau in the top panel and Actin in the bottom panel (A). Quantitation of 2N, 1N and 0N tau normalized to Actin does not display a significant difference between the cohorts (B).
(TIF)

**S4 Fig. Expression of the lysosomal protease Cathepsin D was not significantly different between the younger and the older cohorts in the post-mortem brain tissues.** Representative western blot probed with Cathepsin D and b-actin loading control (A). Quantitation of Cathepsin D normalised to β-Actin displays no significant difference.
(TIF)

**S5 Fig. Unexpressed controls.** Elav Geneswitch (GS) expression system allows for temporal control of pan-neural hTau$^{0N3R}$ expression following introduction of RU486 upon eclosion. No expression is evident when no drug is given (0μm RU486) and a dose dependent increase in tau expression is seen with increasing doses of RU486. 200 μM RU486 gives tau expression that is comparable to that seen with the standard Elav Gal4 so this dose was chosen for all the experiments in this study (A). Atg8 staining in Elav GS/UAS-Atg1, Elav GS/hTau$^{0N3R}$ and Elav GS/htau$^{0N3R;Atg1}$ transgenics after treatment with 200 μM RU486 upon eclosion. During autophagy, Atg8-I is conjugated with a lipid moiety and the lapidated form (Atg8-II) is recruited to the autophagosomal membranes. An elevated Atg8-II level therefore serves as an indicator of autophagy activation. Atg8-II levels are evident in the Atg1 and hTau$^{0N3R}$;Atg1 flies but not to that extent in the hTau$^{0N3R}$ alone flies (B).
(TIF)

**S6 Fig. Impact of hTau expression on endogenous dTau.** Drosophila tau (dTau) was probed using an antibody specific only to dTau (gift from Johnston lab University of Cambridge) and in wild type (wt) OreR flies, dTau levels were found to increase with age (p = 0.0193). This age-related increase was not evident upon Elav Gal4 expression of hTau0N3R. The dTau expression in 6 week wt flies that are expressing hTau0N3R was significantly less than that evident in 6 week wt flies (p = 0,0309) (data is fold change compared to dTau levels in wt flies at 0 week; unpaired two tailed t-test; n = 7–14).
(TIF)

**S7 Fig. Two hypotheses for how autophagy may change with age and influence accumulation of misfolded proteins.** In individuals with a healthy autophagic capacity, autophagic responses increase with age to deal with age-related insults. This may contribute to a decline in soluble phosphorylated tau levels and protect from tau accumulation and sunsequent degeneration in healthy long-lived elderly subjects who do not develop tauopathy (red line) (A). An age-related decline in autophagic capacity is evident in all elderly but in some subjects (blue line) this is not significant enough to encourage tau accumulation and they do not develop

degeneration or tauopathy in other subjects (red line), the age-related impairment occurs to a greater extent and this precipitates accumulation of misfolded proteins and pathogenesis of neurodegenerative diseases like tauopathy.
(TIF)

**S1 Raw images.**
(PDF)

## Acknowledgments

We thank Dr. Anton Page for his electron microscopy training and advice and Ms. Hannah Warming for tissue collection and retrieval.

## Author Contributions

**Conceptualization:** Amritpal Mudher.

**Data curation:** Megan Sealey.

**Formal analysis:** Shreyasi Chatterjee, Megan Sealey.

**Funding acquisition:** Amritpal Mudher.

**Investigation:** Shreyasi Chatterjee, Megan Sealey, Eva Ruiz, Anna Crisford, Emma Luckett, Rebecca Robertson, Philippa Richardson, Paul Grundy, Diederik Bulters, Mariana Vargas-Caballero.

**Methodology:** Shreyasi Chatterjee, Megan Sealey, Eva Ruiz, Keeley Brookes, Sam Green, Anna Crisford, Michael Duque-Vasquez, Emma Luckett, Rebecca Robertson, Philippa Richardson, Girish Vajramani, Paul Grundy, Diederik Bulters, Mariana Vargas-Caballero.

**Project administration:** Shreyasi Chatterjee.

**Resources:** Chrysia M. Pegasiou, Girish Vajramani, Mariana Vargas-Caballero.

**Supervision:** Amritpal Mudher.

**Writing – original draft:** Shreyasi Chatterjee.

**Writing – review & editing:** Christopher Proud, Mariana Vargas-Caballero, Amritpal Mudher.

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
