## [Decision Letter · Decision Letter 0]

29 Nov 2021

PONE-D-21-30304Age-related changes in Tau and Autophagy in human brain in the absence of neurodegenerationPLOS ONE

Dear Dr. Mudher,

Thank you for submitting your manuscript to PLOS ONE. After careful consideration, we feel that it has merit but does not fully meet PLOS ONE’s publication criteria as it currently stands. Therefore, we invite you to submit a revised version of the manuscript that addresses the points raised during the review process.

Reviewers found your study interesting and important. However, they noted several issues that need your attention. You showed that the levels of tau protein were reduced in old samples compared to young ones and concluded that it was resulted from an increase in autophagy. Reviewer suggested that it could be stemmed from the transcriptional repression of tau gene in the old samples. Likewise, reviewer suggested that samples should be analyzed by more anti-phophotau antibodies.  There were several other comments.  

We look forward to receiving your revised manuscript.

Kind regards,

Hemant K. Paudel

Academic Editor

PLOS ONE

Journal Requirements:

4. Please amend the manuscript submission data (via Edit Submission) to include author Sam Green.

5. Please include a copy of Table 1 which you refer to in your text on page 5 and page 9.

Reviewers' comments:

Reviewer's Responses to Questions

**Comments to the Author**

1. Is the manuscript technically sound, and do the data support the conclusions?

Reviewer #1: Partly

Reviewer #2: Yes

2. Has the statistical analysis been performed appropriately and rigorously? 

Reviewer #1: Yes

Reviewer #2: Yes

3. Have the authors made all data underlying the findings in their manuscript fully available?

Reviewer #1: Yes

Reviewer #2: Yes

4. Is the manuscript presented in an intelligible fashion and written in standard English?

Reviewer #1: Yes

Reviewer #2: Yes

5. Review Comments to the Author

Reviewer #1: The authors address these questions by investigating physiological age-related changes in both tau and autophagy in human brain tissues taken from young and old subjects. The results seem to be very important in the research area of neurodegenerative diseases including Alzheimer’s disease. However, I have some points to clarify:

1. The authors showed that the levels of tau protein were reduced in old samples compared to young ones. In this study, they concluded that it was resulted from an increase in autophagy; however, it could be stemmed from the transcriptional repression of tau gene in the old samples. The authors should check mRNA levels of tau using qRT-PCR or RT-PCR.

2. The authors insist that it is unclear whether autophagy contributes to the age-related reduction in soluble tau in the Discussion section (23 page). In autophagy-mediated degradation of tau, lysosomal activity is also critical to its degradation. So, authors did examine the levels of lysosomal enzymes such as ATG9b, cathepsin D, and CLCN7 and so on. To improve I recommend authors to examine those protein levels, at least cathepsin D.

3. Minorly, In the Discussion section, authors described that “With increasing age, a decrease in total tau levels and an increase in expression of two markers of autophagy were evident in the absence of overt neurodegeneration”. The expression seems to give readers a confusion. What are two autophagy markers increased in old samples? LC3II and p62; however, p62 was decreased.

Reviewer #2: This is an interesting and well-conducted study on the relation between total tau in old and young human cortex as well as a model system (Drosophila). However, there are a few issues that have to be addressed and that I list below.

1. On page 19 (of the PDF) the authors state that there is a consensus that there is no age-related decline in autophagy in brain : 1. the authors have to cite various appropriate references backing their statement and 2. I don't think that this consensus really exists as other studies/paper clearly demonstrate a decrease and I would rather recommend to state that this topic is rather controverse and surely this was one reason that the authors performed the current study.

2. Please briefly describe what type of tau hTau0N3R is and what pathology it has in humans.

3. It seems that the authors only used one single phosphorylation site/antibody (PHF1) in their study. I am not sure that, considering the many different sites, analysing just 1 is really sufficient to confirm that tau in their brains is really hyperphosphorylated/pathological. The authors should make a strong case and discuss this issue, possibly as a limitation of their study.

4. Importantly, the authors analyse only total and 1 phospho-tau site, but not aggregated tau which is the most likely cause for neurodegeneration compared to total tau. Autophagy is predominantly known to digest aggregated and misfolded proteins while soluble or oligomeric forms are rather degraded by other protein degradation mechanisms such as the proteasome. How do the authors exclude any influence of the latter on tau levels and why was aggregated tau not analysed? In the discussion decrease of soluble tau is briefly discussed as a speculation and aggregated mentioned, but the different forms of tau have to be explained in the introduction and reasons should be given why aggregated tau which can occur in non-demented older brains, has not been investigated.

5. Kang et al., 2019 (JAD) have demonstrated that autophagy activation participates in the secretion and transfer to neurons from both total and phospho-tau. Please include and discuss this novel observation into your manuscript.

6. 6. On page 1 of the introduction between references 7 and 8 suddenly high numbers (59, 65, 80 etc) appear. Similarly, on the next page suddenly reference 36 appears after 11 which is not correct. These reference numbers have to be integrated and give appropriate numbers as references should be numbered consecutively as they appear in the text.

7. Additionally, "Pegasiou et al., 2020) in the results part needs to be transformed into a number and integrated into the consecutive numbering system.

8. Please explain "eclosion" as it is rather specific for Drosophila and not every reader is an expert in this area.

9. What value does it have to analyse and show just a single case regarding electrophysiology?

10. There is no colour in suppl. fig. 4, please add since otherwise everything looks just grey. If no colour should be used then discriminate the lines/curves differently (using dashed lines, for example.

6. PLOS authors have the option to publish the peer review history of their article (what does this mean?). If published, this will include your full peer review and any attached files.

Reviewer #1: No

Reviewer #2: **Yes: **Gabriele Saretzki

---

## [Author Response · Author response to Decision Letter 0]

19 Jul 2022

Dear Professor Paudel,

We have now responded to the Reviewer comments in the "Response to the Reviewer" document that is attached with this submission. In addition, we have uploaded the Revised manuscript with the track-changes and high-lighted sections as instructed. We have also performed most of the experiments that were suggested by Reviewers 1 and 2. These results have been included as part of the Supplementary information and discussed in this manuscript.

We did submit on time but our manuscript needed a few amendments from the Editorial Office based on PLoS One regulations regarding new author addition and funding information hence the delay.

As of 19-7-22 we have edited the Manuscript 4 times based on the request from a new editor each time.

Thanking You,

Amritpal Mudher

---

## [Decision Letter · Decision Letter 1]

22 Aug 2022

Age-related changes in Tau and Autophagy in human brain in the absence of neurodegeneration

PONE-D-21-30304R1

Dear Dr. Mudher,

We’re pleased to inform you that your manuscript has been judged scientifically suitable for publication and will be formally accepted for publication once it meets all outstanding technical requirements.

Kind regards,

Hemant K. Paudel

Academic Editor

PLOS ONE

Additional Editor Comments (optional):

Reviewers' comments:

Reviewer's Responses to Questions

**Comments to the Author**

1. If the authors have adequately addressed your comments raised in a previous round of review and you feel that this manuscript is now acceptable for publication, you may indicate that here to bypass the “Comments to the Author” section, enter your conflict of interest statement in the “Confidential to Editor” section, and submit your "Accept" recommendation.

Reviewer #1: All comments have been addressed

Reviewer #2: All comments have been addressed

2. Is the manuscript technically sound, and do the data support the conclusions?

Reviewer #1: Yes

Reviewer #2: Yes

3. Has the statistical analysis been performed appropriately and rigorously? 

Reviewer #1: Yes

Reviewer #2: Yes

4. Have the authors made all data underlying the findings in their manuscript fully available?

Reviewer #1: Yes

Reviewer #2: Yes

5. Is the manuscript presented in an intelligible fashion and written in standard English?

Reviewer #1: Yes

Reviewer #2: Yes

6. Review Comments to the Author

Reviewer #1: (No Response)

Reviewer #2: The authors now addressed all outstanding reviewer comments by either performing additional experiments or highlighting limitations and clarifying various study aspects.

7. PLOS authors have the option to publish the peer review history of their article (what does this mean?). If published, this will include your full peer review and any attached files.

Reviewer #1: No

Reviewer #2: **Yes: **Dr. Gabriele Saretzki

---

## [Editor Report · Acceptance letter]

11 Nov 2022

PONE-D-21-30304R1 

Age-related changes in Tau and Autophagy in human brain in the absence of neurodegeneration 

Dear Dr. Mudher:

I'm pleased to inform you that your manuscript has been deemed suitable for publication in PLOS ONE. Congratulations! Your manuscript is now with our production department. 

Kind regards, 

on behalf of

Dr. Hemant K. Paudel 

Academic Editor

PLOS ONE